# RNF213 Loss-of-Function Promotes Angiogenesis of Cerebral Microvascular Endothelial Cells in a Cellular State Dependent Manner

**DOI:** 10.3390/cells12010078

**Published:** 2022-12-24

**Authors:** Vincent Roy, Alyssa Brodeur, Lydia Touzel Deschênes, Nicolas Dupré, François Gros-Louis

**Affiliations:** 1Department of Surgery, Faculty of Medicine, Laval University, Quebec City, QC G1V 0A6, Canada; 2Division of Regenerative Medicine, CHU de Quebec Research Centre, Laval University, Quebec City, QC G1J 1Z4, Canada; 3Department of Neurological Sciences, Faculty of Medicine, Laval University, Quebec City, QC G1V 0A6, Canada

**Keywords:** RNF213, moyamoya disease, angiogenesis, brain microvascular endothelial cells

## Abstract

Enhanced and aberrant angiogenesis is one of the main features of Moyamoya disease (MMD) pathogenesis. The ring finger protein 213 (RNF213) and the variant p.R4810K have been linked with higher risks of MMD and intracranial arterial occlusion development in east Asian populations. The role of RNF213 in diverse aspects of the angiogenic process, such as proliferation, migration and capillary-like formation, is well-known but has been difficult to model in vitro. To evaluate the effect of the *RNF213* MMD-associated gene on the angiogenic activity, we have generated RNF213 knockout in human cerebral microvascular endothelial cells (hCMEC/D3-RNF213^−/−^) using the CRISPR-Cas9 system. Matrigel-based assay and a tri-dimensional (3D) vascularized model using the self-assembly approach of tissue engineering were used to assess the formation of capillary-like structures. Quite interestingly, this innovative in vitro model of MMD recapitulated, for the first time, disease-associated pathophysiological features such as significant increase in angiogenesis in confluent endothelial cells devoid of RNF213 expression. These cells, grown to confluence, also showed a pro-angiogenic signature, i.e., increased secretion of soluble pro-angiogenic factors, that could be eventually used as biomarkers. Interestingly, we demonstrated that that these MMD-associated phenotypes are dependent of the cellular state, as only noted in confluent cells and not in proliferative RNF213-deficient cells.

## 1. Introduction

Moyamoya disease (MMD) is a rare and progressive intracranial arteriopathy where steno-occlusive lesions of the main vessels and branches of the cerebral vasculature lead to transient ischemic attacks or strokes [1,2]. A decrease in cerebral blood flow (CBF) drives the development of fragile and abnormal collateral blood vessels, also called moyamoya vessels (MMV), at the base of the brain to counterbalance the loss of the parenchymal oxygenation [3,4]. Interestingly, early active neovascularization has been identified in pediatric MMD patients, suggesting that this pathophysiological compensatory mechanism may occur before significant ischemic symptoms [5]. The excessive formation of the MMV also correlates with an increase in circulating angiogenic factors, such as vascular endothelial growth factor (VEGF), basic fibroblast growth factor (bFGF) and matrix metalloproteinase 9 (MMP-9) [6,7]. Those factors are found in elevated concentration in plasma or cerebrospinal fluid (CSF) of MMD patients and are known to be secreted by the hypoxic cerebral microenvironment.

MMD occurs worldwide, but its incidence is significantly higher in east Asian populations, such as in Japan, Korea and China [8,9]. Although the etiology of MMD remains undetermined, MMD is known to be associated with genetic predispositions combined with complex environmental factors [10]. Genome-wide and locus-specific association studies identified *RNF213* (p.R4810K variant) as an important susceptibility gene of MMD among east Asian populations [11,12]. The human *RNF213* gene encodes for a large oligomeric protein with two distinct domains: a walker motif with two AAA-type ATPase regions and a ring finger domain [13,14]. The exact mechanism by which mutant RNF213 leads to MMD-associated clinical symptoms remains unknown. Several studies, using in vitro or in vivo models, reported reduced angiogenic activities [15,16,17,18,19,20]. Although interesting, such phenotype may not completely reflect MMD pathophysiological features, where intracranial angiogenesis was shown to be enhanced following the middle cerebral artery stenosis/occlusion [21].

It has been recently shown that the invalidation of the *RNF213* gene in cerebral microvascular endothelial cells (hCMEC) is known to play a crucial role in the maintenance of the blood–brain barrier (BBB) integrity and function [22]. Here, using the same methodological approach, we report the generation of a robust and reproducible in vitro model for the study of MMD, in which loss of RNF213 highly enhanced angiogenesis, one of the main pathological hallmarks of the disease. Additionally, the RNF213-deficient hCMEC/D3 cells also present well-defined pro-angiogenic secretome, which can subsequently promote angiogenesis.

## 2. Materials and Methods

### 2.1. Cell Culture and CRISPR-Cas9-Mediated RNF213 Invalidation

Immortalized human cerebral microvascular endothelial cells (hCMEC/D3) were purchased from Cedarlane Laboratories (Hornby, ON, Canada) and were maintained in a gelatin-coated tissue culture flask in Endothelial Cell Basal Medium MV2 (ECBM-MV2; PromoCell, Heidelberg, Germany) supplemented with 5% fetal bovine serum (FBS; HyClone, Logan, UT, USA), 1.4 µM hydrocortisone (Galenova Inc., St-Hyacinthe, QC, Canada), 5 µg/mL ascorbic acid (Sigma-Aldrich, St. Louis, MO, USA), 1% chemically defined lipid concentrate (Gibco Laboratories, Gaithersburg, MD, USA), 1 ng/mL human basic fibroblast growth factor (bFGF; Sigma-Aldrich), 10 µM HEPES (MP Biomedicals, Santa Ana, CA, USA) and the antibiotics 100 IU/mL penicillin G (Sigma-Aldrich) and 25 μg/mL gentamicin (Sigma-Aldrich). *RNF213* gene knockout in hCMEC/D3 (hCMEC/D3-RNF213^−/−^) was performed using the CRISPR-Cas9 double nicking system as previously described [22]. HCMEC/D3 were cultured in two distinct conditions: at confluency or in a proliferative state. For cells cultured at confluency, hCMEC/D3 were seeded at a density of 6.4 × 10^4^ cells/cm^2^ in culture plates pre-coated with 0.5 µg/mL collagen IV (Sigma) and cultured for 10 days. For cells cultured in a proliferative state, hCMEC/D3 were seeded at a density of 5.2 × 10^2^ cells/cm^2^ and cultured for a maximum of 6 days. Dermal fibroblasts were isolated following breast surgery as previously described [23] and epineural fibroblasts were isolated from the peripheral nerve by an explant method as also mentioned previously [24]. Fibroblasts were cultured in Dulbecco’s Modified Eagle Medium (DMEM, Invitrogen, Burlington, ON, Canada), supplemented with 10% FBS (HyClone) and antibiotics 100 IU/mL penicillin G (Sigma-Aldrich) and 25 μg/mL gentamicin (Sigma-Aldrich). Cell culture plates were kept in an incubator at 37 °C, 8% CO^2^ and 95% relative humidity, and culture media were changed 3 times a week.

### 2.2. Cell Proliferation Assays

Daily population doubling for hCMEC/D3-WT and hCMEC/D3-RNF213^−/−^ was calculated with the formulae previously reported by Cortez Ghio and collaborators [25]. Proliferation rate was also quantitatively analyzed with a standard MTT colorimetric assay. Briefly, cells were seeded into 96-well plates pre-coated with gelatin type A (0.2%, Fisher Scientific, Hampton, NH, USA) and incubated for 1, 2 and 4 days. For each of these time points, cells were incubated with 1 mg/mL 3-(4,5-dimethylthiazol-2-yl)-2,5-diphenyltetrazolium bromide (MTT, Sigma-Aldrich) for 3 h at 37 °C. Purple formazan crystals were dissolved in isopropanol-HCl and absorbance was measured on an automated microplate reader (Bio-Rad, Laboratories Inc, Hercules, CA, USA) at 570 nm.

Cells entering and progressing through the cell cycle were analyzed using the BD BrdU FITC Assay (BD Biosciences, Franklin Lakes, NJ, USA) as described by the manufacturer’s instruction. Briefly, both hCMEC/D3-WT and hCMEC/D3- RNF213^−/−^ in a proliferative state were incubated in presence of 10 μM BrdU for 2 h, trypsinised, fixed, permeabilized and treated with 0.3 mg/mL DNAse I (Sigma-Aldrich). Cells were then labelled with fluorescein isothiocyanate (FITC), anti-BrdU antibody and 7-aminoactinomycin D (7-AAD) for 20 min and 5 min, respectively, and cell division stages were visualized with a BD FACSMelody™ flow cytometer (BD Biosciences). Apoptotic cells were also quantified using the FITC Annexin V Apoptosis Detection kit with 7-AAD (BioLegend, San Diego, CA, USA) according to the protocol. In brief, proliferative hCMEC/D3 were trypsinized, washed and resuspended in Annexin V Binding Buffer. Then, cells were incubated with FITC-conjugated Annexin V and 7-AAD for 15 min and analyzed by flow cytometry.

### 2.3. In Vitro Scratch Assay

WT and mutant hCMEC/D3 were seeded in 12-well plates pre-coated with 0.5 μg/mL collagen type IV (Sigma-Aldrich), cultured until confluency and serum starved overnight. For the scratch test assay, incisions were manually made with micropipette tips as previously described [26] prior standard washing procedure twice with PBS to eliminate cellular debris. Images were taken at different time points from 0 to 48 h after the scratch using an inverted-light microscope. Cellular migration rate was calculated by measuring the closing area and reported as a percentage of the initial scratched area using ImageJ 1.52v software (Wayne Rasband, National Institute of Health (NIH), Bethesda, MD, USA).

### 2.4. Tube Formation Assay

A total of 3 × 10^4^ hCMEC/D3 (WT and RNF213^−/−^) per well, cultured in ECBM-MV2 supplemented with Endothelial Cell GM MV2 pack additives (PromoCell), were seeded in a 96-well plate containing polymerized Matrigel^®^ (Corning Inc., Tewksbury, MA, USA) (50 μL/well) and incubated at 37 °C for 4 h. Cells were then stained with Calcein AM (2 μg/mL; Invitrogen) for 30 min to reveal live cells that formed capillary-like structures. The microvascular networks were imaged using an LSI 700 confocal microscope with Zeiss Axio Imager (Carl Zeiss Microscopy, Jena, Germany) and analyzed using angiogenesis analyzer plugin on ImageJ 1.52v software. To evaluate the effect of the secretome on tube formation, post-confluent hCMEC/D3-conditioned media (WT and RNF213^−/−^) were collected and then incubated with hCMEC/D3-WT as mentioned above.

### 2.5. Microarray Analysis and Bioinformatics

Total RNA was isolated from confluent and proliferative hCMEC/D3-WT and hCMEC/D3- RNF213^−/−^ using RNeasy Mini Kit (QIAGEN, Hilden, Germany) as described by the manufacturer instructions. RNA quality was measured with an RNA 6000 Nano LabChip kit and a 2100 bioanalyzer apparatus (Agilent Technologies, Santa Clara, CA, USA). Microarray-based gene expression characterizations were then achieved using G4851A SurePrint G3 Human GE 8x60K V1 (Agilent Technologies) according to the manufacturer’s protocol and gene linear expression data was extracted with the ArrayStar V12 software (DNASTAR, Madison, WI, USA). The Network Analyst web platform 3.0 [27] was then used to filter out low abundance (5th percentile) and low variance genes (15th percentile) in order to normalize the data using the variance stabilizing normalization method and to perform differential expression and gene ontology enrichment analyses. Datasets were then uploaded and properly analyzed with the Ingenuity Pathway Analysis (IPA) software (QIAGEN, https://www.qiagenbioinformatics.com/products/ingenuitypathway-analysis, 3 November 2022) [28].

### 2.6. Human Angiogenesis Proteome Profiler

The Proteome Profiler Human Angiogenesis Array kit (R&D Systems, Minneapolis, MN, USA) was used to analyse the secreted angiogenic proteins following the manufacturer’s instruction. Briefly, conditioned media were mixed with a cocktail of biotinylated detection antibody and incubated with the array membrane, pre-spotted with specific target proteins, overnight at 4 °C. Captured proteins are visualized using chemiluminescent detection reagents, following a washing and a 30 min incubation time with HRP-conjugated secondary antibody. Immunocaptured signals were recorded using a Fusion Fx7 imager (Vilber Lourmat Sté, France). Spots intensity was measured with ImageJ 1.52v software, normalized with the mean intensity of positive control spots and normalized to the number of cells present in the culture at the moment of conditioned media recovery.

### 2.7. ELISA

The concentrations of VEGF-A from post-confluent hCMEC/D3 conditioned media were quantified by specific ELISA kits (Abcam) according to the manufacturer’s instructions. All samples were run in duplicates and plates were read at a wavelength of 450 nm using a microplate reader (Bio-Rad). The mean concentrations were normalized to the number of cells present in the culture dishes at the moment of conditioned media recovery.

### 2.8. Immunoblotting

Both hCMEC/D3 groups were lysed in 1X RIPA buffer (Abcam, Cambridge, UK) complemented with Pierce^TM^ protease and phosphatase inhibitor mini tablets (Thermo Fisher Scientific, Waltham, MA, USA) and 10 μg/mL DNAse I (Sigma-Aldrich). Total protein concentration was measured with the Bio-Rad Protein Assay (Bio-Rad), equivalent amounts of each sample were then resolved by electrophoresis on polyacrylamide gels and transferred onto a polyvinylidene fluoride blotting membrane (PVDF; Bio-Rad). Membranes were blocked with 5% non-fat dried milk resuspended in Tris-buffered saline (50 mM Tris, 150 mM NaCl, pH 7.5) containing 0.05% Tween-20 (VWR) and incubated overnight at 4 °C with primary antibodies following recommendations by the manufacturers. Antibodies against RNF213 (1:1000; Sigma-Aldrich), AKT (pan) (1:1000; Cell Signaling Technology, Danvers, MA, USA), phospho-AKT (Thr308) (1:500; Cell Signaling), phospho-AKT (Ser473) (1:500; Cell Signaling), ERK1/2 (1:2000; Cell Signaling), phospho-ERK1/2 (Thr202/Tyr204) (1:1000; Cell Signaling), MEK1/2 (1:1000; Cell Signaling), phospho-MEK1/2 (Ser217/221) (1:1000; Cell Signaling), VEGFR2 (1:1000; Cell Signaling) and phospho-VEGFR2 (Tyr1175) (1:1000; Cell Signaling) were used. β-actin (1:2000; Cedarlane, Burlington, ON, Canada) and Vinculin (1:2000; Cell Signaling) were used as loading controls. Goat anti-rabbit IgG or goat anti-mouse IgG peroxidase-conjugated (1:5000; Jackson ImmunoResearch Lab, West Grove, PA, USA) were used as secondary antibodies. Chemiluminescence signals were revealed with Amersham ECL Western Blotting Dection Reagent (GE Healthcare, Little Chalfont, UK), and visualized using the Fusion Fx7 imager (Vilber Lourmat Sté). Signal intensities of each detected band were analyzed using ImageJ 1.52v software, values were normalized to loading control levels and compared to WT control.

### 2.9. Flow Cytometry

VEGFR2 and PECAM-1 expressed on the plasma membrane surface of hCMEC/D3 was evaluated. RNF213-deficient and wild-type cells were detached with trypsin, washed and resuspended in Cell Staining Buffer (BioLegend). Cells were stained with Fixable Viability dye eFluor™ 780 (eBioscience, Santa Clara, CA, USA) for 30 min and then stained with PE anti-human VEGFR2 (BioLegend) and Alexa-488-conjugated anti-PECAM-1 (BioLegend) antibodies for 20 min. Finally, cells were analyzed with a BD FACSMelody™ flow cytometer (BD Biosciences) and data were treated using the FlowJo^TM^ v9 software (Ashland, OR, USA).

### 2.10. Tri-Dimensional Capillary Network Formation

Epineural or dermal fibroblasts were seeded at a density of 1.5 × 10^4^ cells/cm^2^ in 12-well plates with paper anchors and cultured in complete DMEM medium containing 50 μg/mL ascorbic acid (Sigma-Aldrich) to form manipulable fibroblast sheets [29]. After 3 weeks of culture, hCMEC/D3 were seeded on two thirds of the generate fibroblast sheets at density of 3.3 × 10^4^ cells/cm^2^ and cultured for 1 extra week in complete DMEM/ECBM-MV2 (1:1 ratio) containing 50 μg/mL ascorbic acid. Three cell sheets were stacked and clipped together, and 3D cellular sheets were cultured for 7 additional days. Tri-dimensional (3D) vascularized cell sheets were then raised at the air-liquid interphase and cultured for 10 more days in complete DMEM containing 50 μg/mL ascorbic acid.

### 2.11. Immunofluorescence

HCMEC/D3 and 3D vascularized cell sheets were fixed in 4% paraformaldehyde (PFA) (Electron Microscopy Sciences) for 30 min at 4 °C, washed and then blocked with PBS containing 5% goat serum (Invitrogen) and 0.3% Triton-X100 (Bio-Rad) for 1 h at room temperature. Cells morphology and cytoskeleton organization were determined by a direct immunostaining of F-actin with rhodamine labelled phalloidin (10 μg/mL; Sigma-Aldrich). For the 3D microcapillaries network analysis, specific primary antibodies against PECAM-1 (1:100; R&D Systems), VEGFR2 (1:100; Sigma-Aldrich) and Chondroitin Sulfate (NG2; 1:400; BD Biosciences, San Jose, CA, USA) were incubated with whole construct overnight at 4 °C. 3D vascularized constructs were washed and incubated for 2h at room temperature with Alexa Fluor 633-labelled donkey anti-sheep IgG (1:500; Invitrogen) for PECAM-1, Alexa Fluor 594-labelled goat anti-rabbit IgG (1:500; Invitrogen) for VEGFR2 and Alexa Fluor 488-labelled goat anti-mouse IgG (1:500; Invitrogen) for NG2. Nuclei were counterstained with Hoechst 33258 (Sigma-Aldrich). Images were visualized using an LSI 700 confocal microscope with Zeiss Axio Imager (Carl Zeiss Microscopy).

### 2.12. Statistical Analysis

Comparisons between different experimental groups were performed with GraphPad Prism 9 software using unpaired two-tailed Student’s *t*-tests with Welch’s correction or one-way ANOVA with Tukey’s multiple comparison tests. Results are reported as average values ± standard deviation and differences were considered significant when *p*-value ≤ 0.05. Statistical significance is represented by asterisks.

## 3. Results

### 3.1. RNF213 Regulates Proliferation and Cell Cycle of Cerebral Endothelial Cells

We first wanted to confirm RNF213 protein expression and to investigate cellular morphology in RNF213-deficient hCMEC/D3 generated using the CRISPR-Cas9 gene editing system. This technique allowed us to efficiently knockout the expression of RNF213 (*p* < 0.01) (Figure 1a). CRISPR-mediated RNF213 knockout drastically altered cerebral endothelial cell morphology, where F-actin immunostaining revealed more elongated cells (Appendix A). The MTT proliferation assay indicated that RNF213-deficient hCMEC/D3 led to a significant reduction in cell proliferative rate (*p* < 0.0001) (Figure 1b). More specifically, the daily population doubling was decreased by approximately 40% (*p* < 0.0001) (Figure 1c). Cell cycle flow cytometry analyses, using a BrdU assay, indicated that RNF213 invalidation led to a significant increase in the proportion of hCMEC/D3 in G1-phase (51.8 ± 3.0%) when compared with hCMEC/D3-WT (34.0 ± 3.0%; *p* < 0.0001) (Figure 1d,e and Appendix A). In contrast, proportions of hCMEC/D3-WT in S-phase (43.0 ± 3.2%; *p* < 0.0001) and G2+M-phase (21.5 ± 3.7%; *p* < 0.05) were significantly decreased in comparison to RNF213-deficient hCMEC/D3 (27.9 ± 3.6% and 17.3 ± 2.7%, respectively). We also evaluated early and late apoptosis events in hCMEC/D3-RNF213^−/−^ and found no differences in both groups (Figure 1f).

### 3.2. CRISPR/Cas9-Mediated RNF213 Knockout Drives Cellular Migration as Well as the Angiogenic Activity of Cerebral Endothelial Cells

To test whether RNF213 is involved in the angiogenic process of hCMEC/D3, we first assessed endothelial cell migration, essential to angiogenesis, using a standard in vitro scratch assay. CRISPR-mediated RNF213 knockout significantly enhanced cellular migration of hCMEC/D3 as early as 8 h following scratches (*p* < 0.001) (Figure 2a,b). Tube formation assays were also performed in order to reveal the effect of stable RNF213-deficient hCMED/D3 on angiogenesis. As opposed to what was previously reported in the literature, we observed here that functional loss of RNF213 rapidly enhanced tube-like structure formation (Figure 2c). Indeed, the total microvessel network length, calculated using the angiogenesis analyzer plug-in of ImageJ, formed by RNF213-deficient hCMEC/D3, was 1.5-fold (*p* < 0.0001) greater than the wild type controls (Figure 2d). In addition, other indexed angiogenesis parameters such as the number of branches, master segments and master junctions were also measured and found to be significantly increased in the RNF213 knockout group (Appendix A).

### 3.3. Gene Expression and Functionally Enriched Analyses in RNF213-Decficient hCMEC/D3 in A Proliferative or Confluent State

To delineate the molecular effect of RNF213 depletion in brain endothelial cells, a microarray analysis was conducted in cells cultured at two distinct cellular states, i.e., cells in proliferation and cells at confluency. Principal component analysis of gene expression profiles revealed lineage-specific and state-specific segregations for both groups (Appendix A). Differentially expressed genes (DEGs) were identified using the Network Analyst website and were considered significant with fold change of >2 and an adjusted *p*-value < 0.05. For RNF213-deficient hCMEC/D3 in proliferation, a total of 1575 DEGs between wild-type and RNF213 knockout cells, from which 720 genes were up-regulated and 855 down-regulated (Figure 3a). In contrast, 1713 DEGs were detected in RNF213-deficient hCMEC/D3 cultured at confluency, from which 913 genes were up-regulated in hCMEC/D3-RNF213^−/−^ and 800 were down-regulated (Figure 3b). The detailed information on DEGs for both conditions is listed in Appendix A. When comparing the common DEGs associated to the culture state, 490 genes were decreased (Figure 3c) and 515 were increased in RNF213 knockout endothelial cells (Figure 3d).

Notably, gene expression profiles were also found to be modulated according to the tested cellular states, in proliferation and at confluency, in both wild-type and RNF213-deficient cells. Actually, 585 DEGs were detected in hCMEC/D3-RNF213^−/−^, from which 348 genes were up-regulated and 237 were down-regulated (Appendix A), whereas 244 DEGs were identified in hCMEC/D3-WT, from which 104 were increased and 140 were decreased (Appendix A). Interestingly, only 30 DEGs (up- and down-regulated) were found to be shared in both cell lineages (Appendix A).

Afterward, gene ontology (GO) enrichment analyses were achieved in order to identify affected biological processes in both proliferative and confluent RNF213-deficient hCMEC/D3. In proliferative RNF213-deficient cells, biological processes were positively associated with gliogenesis and regulation of Rho protein signal, GTPase activity, Ras protein signal and MAP kinase activity (Figure 3e), whereas they were negatively associated with angiogenesis, vasculature development, cell migration, cell proliferation and focal adhesion assembly (Figure 3e). In contrast and of particular interest, phototransduction, vasculature development and regulation of MAPK cascade, cytokine biosynthesis and cell differentiation were found to be enhanced in confluent RNF213-deficient cells (Figure 3f). Down-regulated biological processes such as focal adhesion assembly, regulation of actin filament length, cytoskeleton organization, cell adhesion and gliogenesis were also identified in confluent RNF213 knockout cells (Figure 3f). Overall, our findings unveil that the invalidation of RNF213 in brain microvascular endothelial cells greatly affected gene expression profiles, which were also modulated according to the state of the cell culture. 

### 3.4. Inactivation of RNF213 Promotes Angiogenesis through Activation of VEGFR2

Next, we analyzed the content of conditioned culture media from confluent RNF213-deficient hCMEC/D3 by quantifying the pro-angiogenic factors present in the supernatants using an angiogenesis proteome profiler immunoassay. Interestingly, an overall pro-angiogenic signature in RNF213-deficient hCMEC/D3 conditioned media was detected (Figure 4a and Appendix A). Several soluble proteins such as amphiregulin, DPP4, GM-CSF, IL-8, MMP-9 and VEGF were found to be overrepresented in the RNF213-deficient hCMEC/D3 conditioned media when analyzed at confluency (Figure 4a and Appendix A). No significant variations were observed between both groups (WT and RNF213^−/−^) in proliferative hCMEC/D3 conditioned media (Appendix A). VEGF-A protein overexpression, known to be an important pro-angiogenic protein in MMD, was also confirmed by ELISA using specific VEGF-A commercially available detection kits (Figure 4b) [4]. Indeed, we measured a significant increase of VEGF-A (2148 ± 1258 pg/mL; *p* < 0.01) compared to the control cells (377.8 ± 216.0 pg/mL) (Figure 4b). Notably, CRISPR-Cas9-mediated RNF213 knockout also led to a 5.2-fold increase in phosphorylation level of VEGFR2 (*p* < 0.05), while having no influence on the total VEGFR2 protein expression (Figure 4c,d). Flow cytometry analyses have shown a clear increase in VEGFR2 expression, as well as a reduction of PECAM-1, at the plasma membrane surface of hCMEC/D3^−/−^ (Appendix A). We also assessed, using the relevant tube-formation biological assay, the angiogenic activity of RNF213-deficient hCMEC/D3 conditioned media and found a significant enhancement of tube-like formation in vitro (Figure 4e). Actually, a 2.9-fold increase of the microvessel network length was quantified (*p* < 0.001) using RNF213-deficient hCMEC/D3 conditioned media (Figure 4f). Moreover, the number of nodes, branches, segments, master segments, junctions and master junctions were also significantly increased (Appendix A).

### 3.5. VEGF Signaling Pathway Prediction Analysis of Functions Associated with Angiogenesis

As previously shown by western blot analysis (Figure 4c,d), the phosphorylation of VEGFR2 (Y1155) was found to be increased, which indicated that the VEGF signaling pathway might also be differentially modulated. We thus subsequently performed an in-depth analysis of functionally enriched pathways using microarray data generated from RNF213-deficient hCMEC/D3 (Figure 5). First, angiogenesis was predicted to be activated in confluent cells, while predicted to be inhibited in proliferative cells confirming our previous results (Figure 5 and Appendix A). Indeed, VEGFA, a strong regulator of angiogenesis, was found to be upregulated in confluent RNF213-deficient hCMEC/D3, whereas downregulated in proliferative cells (Figure 5). In contrast, VEGFB and VEGFC were both upregulated in the two groups. Meanwhile, endothelial cell proliferation was predicted to be activated in proliferative cells but inhibited in confluent cells (Figure 5 and Appendix A). As hCMEC/D3-RNF213^−/−^ has a reduced proliferation rate (Figure 1b,c), we assessed the activation level of various signaling pathways associated with cell proliferation, such as the PI3K/Akt and the Ras/MAPK pathways. Overall, activation by phosphorylation was similar between the groups (Appendix A). Interestingly, a 1.9-fold increase of total Akt (*p* < 0.001) was measured in RNF213-deficient cells (Appendix A). While no differences were observed for MEK1/2, a significant 1.6-fold upregulation was measured for the total ERK1/2 protein levels (*p* < 0.01) (Appendix A). Finally, endothelial cells migration was predicted to be inhibited in both groups (Appendix A), while an increase was observed with the in vitro scratch assay (Figure 2a,b).

### 3.6. Inactivation of RNF213 Promotes the Formation of Abnormal Microvascular Brain Endothelial Cell Network in Tridimensional Tissue-Engineered Cell Culture Model

Ultimately, we used tissue engineering to recreate an advanced 3D local cellular microenvironment, containing endogenous extracellular matrix proteins, shown to have biochemical/biophysical properties closer to native tissue [29,30,31]. Thereafter, brain endothelial cells were seeded on the 3D exogenous-free material tissue-engineered cellular sheets allowing the formation of a well-defined capillary network and structure (Figure 6a). Interestingly, RNF213-deficient hCMEC/D3 formed a tiny, dense and anarchic VEGFR2-postive/PECAM-1-negative microvessel network in comparison to hCMED/D3-WT. Both conditions led to structured capillary-like network, which were surrounded by NG2-positive pericytes known to be produced from advanced differentiation of fibroblasts [32] (Figure 6b). Note that NG2-positive cells, detected in tissue-engineered cell sheets made of RNF213-deficient hCMED/D3, were unorganized and more randomly distributed. Interestingly, RNF213-deficient hCMEC/D3 seeded into tissue-engineered constructs made with epineural fibroblasts formed thinner vessels and better mimic intracranial microvasculature (Appendix A).

## 4. Discussion

The pathogenesis of MMD associated with RNF213 still remains unclear. Mice lacking Rnf213, or genetically modified transgenic mice developed normally with no pathological symptoms reminiscent to MMD [18,20,33]. However, chronic hind-limb ischemia induction in RNF213 mutant mice triggered formation of predominant collaterals in the ischemic leg [21,34]. Interestingly, Rnf213 knockdown in zebrafish showed abnormal development of craniocervical blood microvessels [11]. Yet, these studies suggest the existence of interspecies differences for RNF213 dysfunction and possible compensatory pathways in those animals. Several other in vitro models, describing divergent and disease irrelevant data, have also been described to better depict the role of RNF213 [15,16,17]. However, the establishment of robust human-based RNF213^−/−^ cellular models have largely failed to reveal meaningful MMD-associated pathophysiological mechanisms. A potential explanation for this discrepancy has been the use of inappropriate endothelial cell lines, such as human umbilical vein endothelial cells (HUVEC) and human coronary artery endothelial cells, to study complex cerebrovascular diseases. Indeed, it has been shown that the origin of different microvascular endothelial cells isolated from various anatomical locations display distinct gene expression patterns that parallel phenotypic characteristics [35,36]. Of particular interest, cerebral microvascular endothelial cells uniquely formed tighter capillary endothelium and inter-endothelial tight junctions [37], which are known to play a crucial role in the maintenance of the BBB integrity and function [22]. It is therefore likely that human brain microvascular cells will display reminiscent angiogenic properties and would be a better model to study cerebrovascular diseases such as MMD. Concomitantly, those discrepancies can also be attributed to transient RNF213 knockdown expression, using interferent RNA or expression plasmid, which were reported in several studies.

The establishment of robust human-based RNF213 knockout cellular models have largely failed to reveal meaningful MMD-associated pathophysiological phenotypes. In the present study, we have used the CRISPR-Cas9 double nickase gene-editing technique to invalidate RNF213 in well-characterized immortalized human cerebral microvascular endothelial cells, which were previously shown to predominantly express RNF213 [22]. Here, we further characterized this CRISPR-mediated RNF213-deficient model and showed that RNF213 acted as an anti-angiogenic factor in brain microvascular endothelial cells in vitro. For instance, it has recently been demonstrated by Zhang and collaborators that RNF213 loss-of-function is associated with an increase of endothelial cells microvessel network formation [38]. We also delineate the contribution of the VEGF-A/VEGFR2 signaling cascades in the pro-angiogenic activity of post-confluent RNF213-deficient cells. In-depth transcriptional analyses revealed that angiogenesis is predicted to be strongly activated in confluent RNF213-deficient cells, while it is predicted to be inhibited in proliferative endothelial cells. Previous reports demonstrated such modulation in gene expression profiles according to the cell culture density, denoting that cellular state may be a critical factor and therefore must be considered when analyzing transcriptomic data [39]. This might also explain why several described RNF213 in vitro models showed divergent outcomes [15,16,17]. In addition, we showed that the confluent RNF213-deficient hCMEC/D3 secretome contained a collection of pro-angiogenic factors, such as VEGF-A. These findings suggest that the secretion of those factors by RNF213-deficient brain endothelial cells, delivered into the intercellular space, is likely to promote pathogenic angiogenesis associated with MMD.

VEGF-A and its receptors VEGFR-2 exert major roles in physiological as well as pathological angiogenesis through distinct signal transduction pathways regulating proliferation and migration of endothelial cells [40]. Our results demonstrated that RNF213 loss-of-function induced phosphorylation of VEGFR2 on Tyr1175 and immobilization of the receptor at the plasma membrane, suggesting a possible role of VEGFR2 signaling in pathological angiogenesis in MMD. In fact, the phosphorylation on Tyr1175 is essential for the activation of the receptor and is known to regulate permeability, proliferation and migration of vascular endothelial cells [41,42]. Likewise, phosphorylation of VEGFR2 on Tyr1175 promotes the activation of Akt and ERK1/2 kinases, also through phosphorylation, which then promotes cell proliferation and migration [43,44]. However, a clear decline in the rate of cell division and an alteration in the endothelial cell morphology of RNF213-deficient hCMEC/D3 were observed in this study as also previously reported [22]. Remarkably, an upregulation of Akt and ERK1/2 protein levels has been observed in our model, suggesting a possible compensatory mechanism to bypass this reduction of cellular proliferation shown to differentially affect specific cell cycle phases. Ohkubo and his collaborators have demonstrated that silencing RNF213 in HUVECs also reduced the proliferation rate and altered the cell cycle phase [17]. Although we observed both an increase in cells in G1 phase and a decrease in S phase, in our studies, less endothelial cells were in G2 and mitosis phase. They then notice a significant decrease in Akt phosphorylation upon RNF213 silencing, which was not demonstrated by our data using our method of invalidation [17].These differences could be explained by the origin of endothelial cells or by the approach to reduce/invalidate RNF213 expression. Functions prediction analysis of our microarray dataset revealed that cellular state particularly affected genes expression associated with endothelial cell proliferation. Indeed, confluent RNF213-deficient hCMEC/D3 were predicted to be less proliferative, while proliferative cells were predicted to be in active mode transcriptionally, even if a reduction in the daily doubling was measured.

It has been previously shown that CRISPR-Cas9-mediated RNF213 invalidation in hCMEC/D3 significantly reduced PECAM-1 expression and its glycosylation [22]. PECAM-1 is known to be involved in angiogenesis and play specific roles in atherogenesis and collateral blood vessels formation [45,46,47]. During angiogenesis, PECAM-1 also acts independently by contributing to the adhesion process and the intracellular signalization that is needed for cell motility and the proper organization of capillaries [46]. Undeniably, GO terms analysis has revealed that RNF213 loss-of-function in hCMEC/D3 negatively affected the regulation of cell migration and the focal adhesion assembly and cell adhesion, which are also functions involved in cell migration processes. 

It is also known that PECAM-1 glycosylation regulates VEGFR2-dependant signaling through formation and stabilization of PECAM-1/VEGFR2/β3 integrin proteins complex [48,49]. On one hand, perturbation of this complex, by knocking-out specific N-glycosylation-associated enzyme, may cause internalization of the VEGFR2 receptor and subsequent activation of the angiogenic process [50]. Perturbation of these complexes did not enhance the intracellular death signaling pathway, which was previously reported by others [50]. In general, although VEGFR2 internalization events occur in endothelial cells, it is not necessary for a proper activation of the downstream intracellular signaling [41,42]. While a reduction of PECAM-1, and its glycosylation state, was associated with RNF213 invalidation [22], the angiogenic process was clearly enhanced in our cellular model. One explanation would be that less PECAM-1, accompanied by more activated VEGFR2 at the surface of RNF213-deficient brain endothelial cells, may cause aberrant angiogenesis. Intracranial bleeding following the rupture of the MMV is a rare but well-known clinical manifestation found in MMD patients with advance stage of the pathology [51,52,53] that may be associated with PECAM-1/VEGFR2 and VEGFA signaling.

Of particular interest, the inosculation of RNF213-deficient hCMEC/D3 within 3D fibroblast ECM self-secreted constructs [22] allowed the formation of aberrant microcapillary-like networks, similar to the compensatory MMV occurring upon the stenosis/occlusion of the cerebral blood artery. Compelling evidence from recent studies demonstrated that fibroblasts originating from different tissues display different patterns of gene expression [35,54]. Here, we used epineural fibroblasts to recreate a 3D scaffold for microvessels formation, instead of skin fibroblasts, and generated a microenvironment which could better mimic a native cerebral/neuronal tissue. Although RNF213-deficient hCMEC/D3 clearly formed tiny and anarchic microcapillaries, the positioning of NG2-positive pericytes is also indicative of an immature network [55]. Interestingly, it has been previously demonstrated that pericytes can be directly differentiated from fibroblasts in a tissue-engineering skin model [32]. Indeed, complexification of our vascularized 3D model by incorporating additional cell types, such as glial cells, to better mimic the brain microenvironment may be of particular significance in future studies for the characterization of RNF213-associated MMD features in vitro.

Lastly, transcriptomic analyses also revealed that gliogenesis, the generation of glial cells, is predicted to be inhibited in confluent RNF213-deficient hCMEC/D3, while it is predicted to be activated in proliferative cells. Due to its association with carcinogenesis, lower RNF213 expression was recently reported to decrease patient survival rates in glioblastoma cases [56]. Glial cells are non-neuronal cell types found within the nervous system and are also typically a part of the neurovascular unit and components of the BBB [57]. This result further supports a role for RNF213 in the formation and maintenance of the BBB as previously reported [22].

## 5. Conclusions

In brief, our study brings novel piece of information linking angiogenesis and collateral vessel formation in RNF213- and MMD-associated pathogenesis. Combining CRISPR-Cas9-mediated loss-of-function of RNF213 in brain endothelial cells and 3D cellular cultures, we were able to generate a reliable MMD in vitro RNF213-deficient model demonstrating reminiscent and characteristic pathological disease manifestations such as enhanced and aberrant angiogenesis. The results presented in this study indicate that VEGFR2-targeted therapy may be considered in the treatment of MMD. The RNF213-mediated knockout model presented here could become of particular importance to the study of angiogenesis and vascular integrity in MMD and for preclinical drug screening applications. In the future, generation of CRISPR-mediated RNF213 point mutation- or patient-derived 3D models, incorporating patient- or iPSC-derived endothelial cells carrying specific point mutations in the *RNF213* gene, will help us to better understand MMD pathogenesis and other RNF213-associated vascular abnormalities. Such personalized models could then become valuable tools for better studying MMD; they would provide data-driven insights into disease mechanisms and evolution, predict responses to therapy and inform personalized treatment strategies.

## Figures and Tables

**Figure 1 cells-12-00078-f001:**
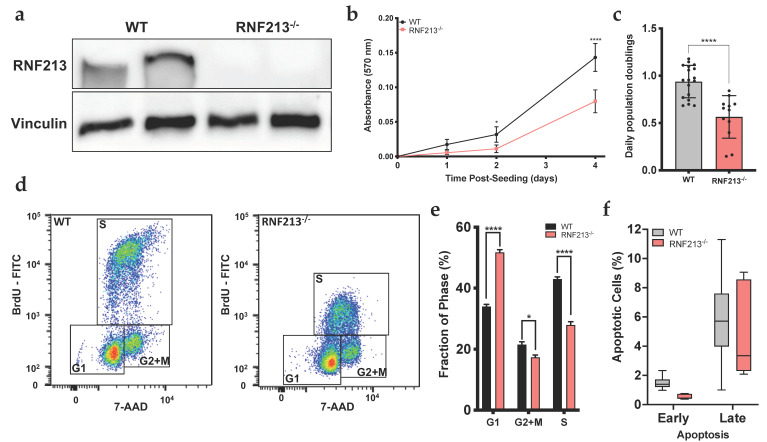
RNF213 invalidation and its effect on cell proliferation and cell-cycle regulation. (**a**) Western blot analysis showing the expression level of RNF213. Vinculin was used as a loading control. Total protein was extracted from confluent cells; (**b**) MTT assay over time to evaluate cell proliferation. *n* = 9–12; (**c**) Measurement of proliferation rate by calculating the daily doubling population. *n* = 12; (**d**) Cell-cycle analysis of proliferative hCMEC/D3 by flow cytometry using the BrdU FITC assay and 7-AAD labelling. *n* = 9; (**e**) Quantification of the proportion of cells in G1, G2+Mitose and S phases from the cell-cycle analysis; (**f**) Early and late apoptosis of hCMEC/D3 measured by flux cytometry analysis. Data from proliferative cells are shown. *n* = 6–9. * *p* < 0.05, **** *p* < 0.0001.

**Figure 2 cells-12-00078-f002:**
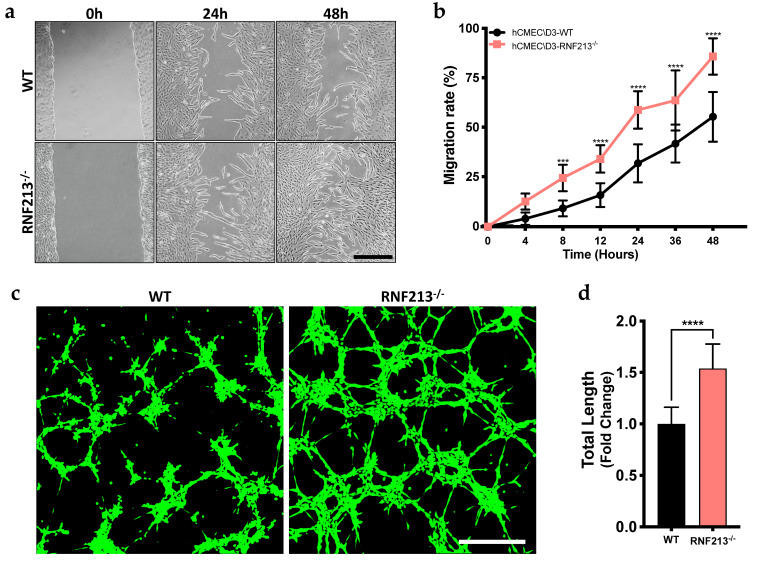
RNF213 deficiency promotes cell migration and tube-like formation. (**a**,**b**) In vitro scratch assay to evaluate hCMEC/D3 migration rate. Images were captured at different time points using phase-contrast microscope and the rate of migration was measured by reporting the total scratch area at each point to the area at time 0 h. *n* = 6–7; (**c**) Tube formation assay on Matrigel^®^ with hCMEC/D3; (**d**) Quantification of the capillary-like networks total length measured using angiogenesis analyser on ImageJ software and reported in fold change relative to WT control. *n* = 10. Scale bar = 500 μm. *** *p* < 0.001 and **** *p* < 0.0001.

**Figure 3 cells-12-00078-f003:**
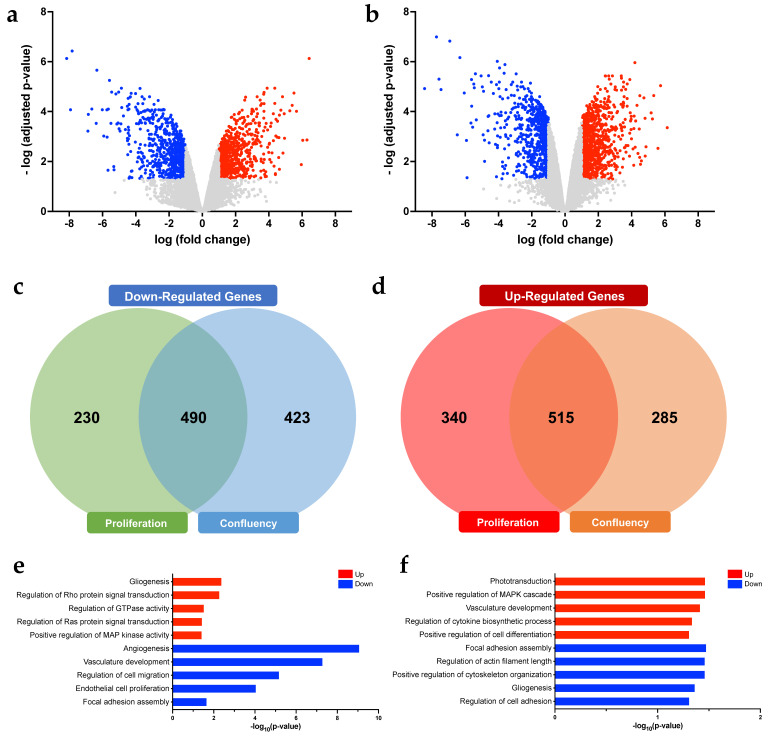
Determination of differentially modulated genes and enriched biological processes in proliferative and confluent RNF213-deficient hCMEC/D3. Volcano plots showing the DEGs in RNF213 knockout endothelial cells cultured in proliferation (**a**) or at confluency (**b**). Venn diagrams of significantly down-regulated (**c**) and up-regulated genes (**d**) in function of the cellular state of RNF213 knockout endothelial cells. Enriched GO terms of biological processes of interest regulated in proliferative (**e**) or confluent (**f**) RNF213-deficient hCMEC/D3. Investigations were all performed using the Network Analyst platform. Significantly enriched GO terms are shown with Benjamini-Hochberg FDR-corrected adjusted *p*-values. *n* = 4/groups.

**Figure 4 cells-12-00078-f004:**
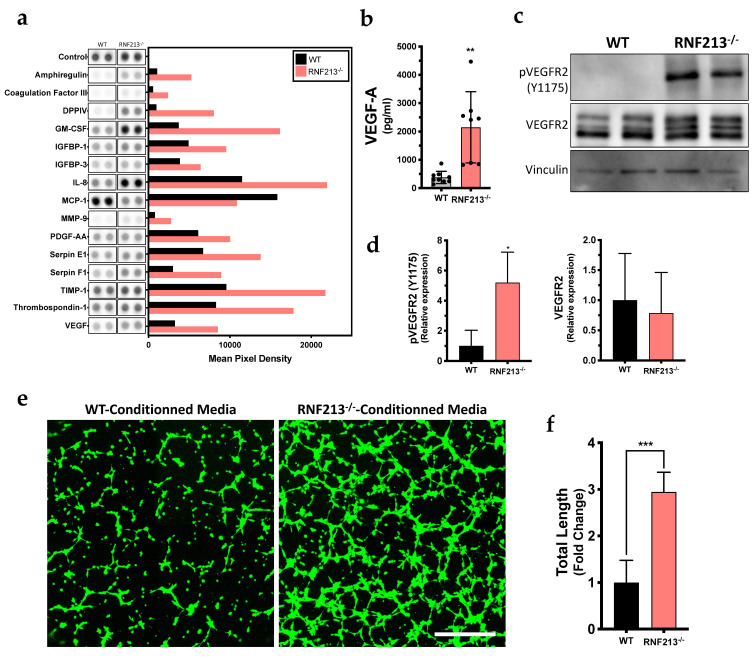
RNF213 invalidation enhanced the secretion of pro-angiogenic factors. (**a**) Profiling of the angiogenic secretome of hCMEC/D3 monolayer using the human angiogenesis proteome arrays. The duplicate spots corresponding to the most altered proteins are shown. Mean pixel intensity was normalized to the number of cells present in the well at the moment of the supernatant recovery. *n* = 2; (**b**) ELISA for secreted VEGF-A. Total proteins concentration is shown in pg/mL of conditioned media and are normalized to the number of cells. *n* = 8; (**c**) Western blot analysis of the receptor VEGFR2 and the phosphorylation levels of Tyr 1175 in hCMEC/D3-RNF213^−/−^ and the WT control cultured at confluency. Vinculin was used as a loading control; (**d**) Densitometry analysis of the total VEGFR2 and the phosphorylation levels. Total VEGFR2 levels are normalized to β-actin and shown to the relative intensity of the WT control. The phosphorylated form of VEGFR2 was normalized to total VEGFR2. *n* = 4; (**e**) Tube formation assay on Matrigel^®^ with conditioned media from hCMEC/D3 monolayer on hCMEC/D3-WT; (**f**) Quantification of the capillary-like networks total length reported in fold change relative to the control with WT conditioned media. *n* = 10. Scale bar = 500 μm. * *p* < 0.05, ** *p* < 0.01 and *** *p* < 0.001.

**Figure 5 cells-12-00078-f005:**
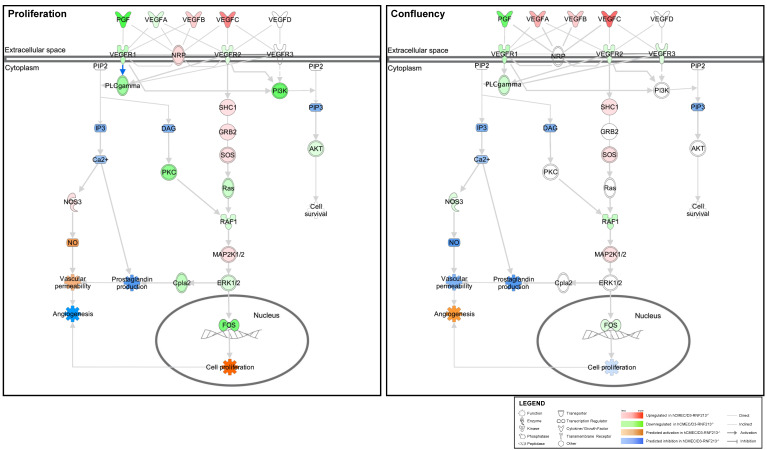
Enrichment of VEGF signaling pathway by the IPA software in proliferative and confluent hCMEC/D3 invalidated in RNF213. The signaling networks were generated with normalized genes expression values, which predict biological functions such as angiogenesis and cell proliferation.

**Figure 6 cells-12-00078-f006:**
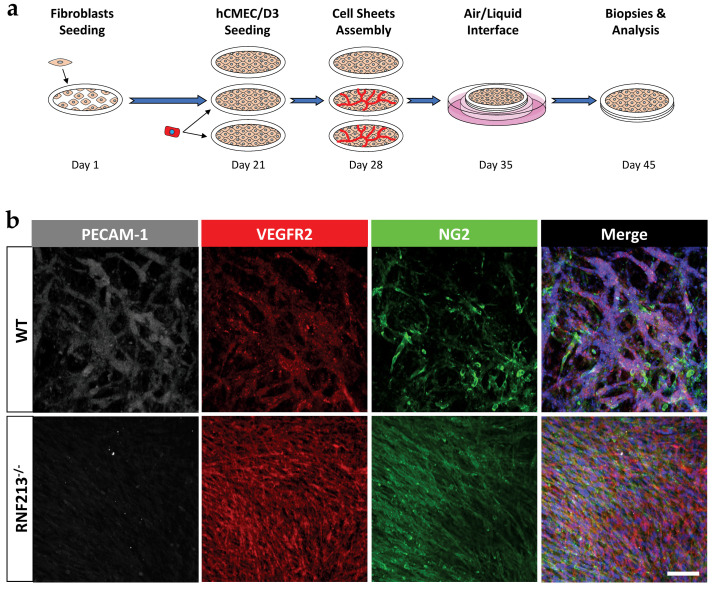
RNF213 deficiency is associated with the formation of moyamoya-like vessels in vitro. (**a**) Timeline of the experimental procedure to generate 3D vascularized constructs using the self-assembly method of tissue engineering. Briefly, epineural fibroblasts were seeded and cultured for 21 days in presence of ascorbic acid. HCMEC/D3 were then added on the second and third cell sheets and cultured in a submerged condition for 7 days. Three cell sheets were stacked, cultured for an additional 7 days and raised at air-liquid interface for 10 days; (**b**) Visualization of the 3D microvascular networks by immunofluorescence staining for PECAM-1 (grey) and VEGFR2 (red) that were imaged by confocal microscopy. Pericyte-like cells were immunostained with NG2 (green). Nuclei were stained with Hoechst (blue). Scale bar = 100 μm.

## Data Availability

Not applicable.

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
