# Peer review of "RNF213 Loss-of-Function Promotes Angiogenesis of Cerebral Microvascular Endothelial Cells in a Cellular State Dependent Manner"

_cells, 2022, doi:10.3390/cells12010078_

Round 1
Reviewer 1 Report
The authors have generated a RNF213 k/o model in human cerebral microvascular endothelial cells using CRISPR-Cas9 and evaluated the effect of the RNF213 MMD-associated gene on angiogenic activity. They have demonstrated that their model shares pathogenic characteristics with MMD. They also showed a differential transcriptome profile associated with cell states. In vitro models to study the angiogenic processes associated with MMD are lacking. The work described in this manuscript is highly relevant. The authors have clearly stated the problematic and the results are supporting their research objectives. The manuscript is well-written.
Comments:
· Figure 3 c and d, the Venn diagram are not showing. Might be a problem with the download but should be verified.
· In section 3.5, it is mentioned “As previously observed, the VEGF signalling was differentially modulated”. It was not clear if the authors are referring to the microarray expression data or the proteome analysis. If it is referring to the microarray, the dysregulation of the VEGF signaling pathway should be emphasis. Are genes of this pathways included in the angiogenesis pathway?
· The discussion should be improved by adding information on the future use of this model. The authors mention in the conclusion “In the future, generation of CRISPR-mediated RNF213 point mutation- or patient-derived models”. What the model they are describing in this manuscript? Is it just a proof-of-principle to produce patient-derived models or could it be use to enhance studies of disease mechanisms and treatment?
Author Response
We thank the Reviewers for their positive and insightful comments on this manuscript.
-The authors have generated a RNF213 k/o model in human cerebral microvascular endothelial cells using CRISPR-Cas9 and evaluated the effect of the RNF213 MMD-associated gene on angiogenic activity.
-They have demonstrated that their model shares pathogenic characteristics with MMD.
-They also showed a differential transcriptome profile associated with cell states.
-The work described in this manuscript is highly relevant.
-The authors have clearly stated the problematic and the results are supporting their research objectives.
-The manuscript is well-written.
Comments:
- - “Figure 3 c and d, the Venn diagram are not showing. Might be a problem with the download but should be verified.”
We are truly sorry about that; a problem during the initial figure upload occurred for an unknown reason. We will make sure this problem will not occur while submitting the revised version of the manuscript.
- - “In section 3.5, it is mentioned “As previously observed, the VEGF signalling was differentially modulated”. It was not clear if the authors are referring to the microarray expression data or the proteome analysis. If it is referring to the microarray, the dysregulation of the VEGF signalling pathway should be emphasis. Are genes of this pathways included in the angiogenesis pathway?”
We agree with the reviewer the confusion regarding this section. We rephrased the confusing sentences, which can now be read as: “As previously shown by western blot analysis (Figure 4C-D), the phosphorylation of VEGFR2 (Y1155) was found to be increased indication that the VEGF signalling pathway might also be differentially modulated. We thus subsequently performed an in-depth analysis of functionally enriched pathways using microarray data generated from RNF213-deficient hCMEC/D3”.
- - “The discussion should be improved by adding information on the future use of this model. The authors mention in the conclusion “In the future, generation of CRISPR-mediated RNF213 point mutation- or patient-derived models”. What the model they are describing in this manuscript? Is it just a proof-of-principle to produce patient-derived models or could it be use to enhance studies of disease mechanisms and treatment?”
Thanks for bringing this point. The discussion has been bonified as suggested: “The RNF213-mediated KO model, presented here, could become of particular importance to study angiogenesis and vascular integrity in MMD and for preclinical drug screening applications. In the future, generation of CRISPR-mediated RNF213 point mutation- or patient-derived 3D models, incorporating patient- or iPSC-derived endothelial cells carrying specific point mutations in the RNF213 gene, will help to better understand MMD pathogenesis and other RNF213-associated vascular abnormalities. Such personalised models could then be a valuable tool to better study MMD, to provide data-driven insight into disease mechanisms and evolution, to predict response to therapy and to inform personalized treatment strategies.”

Reviewer 2 Report
Manuscript title: RNF213 loss-of-function promotes angiogenesis of cerebral microvascular endothelial cells in a cellular state dependent manner
In this manuscript, authors generated an in vitro model of Moyamoya disease (MMD) using the CRISPR-Cas9 system. In this model, knockout of RNF213 promotes angiogenesis of cerebral microvascular endothelial cells in a cellular state dependent manner, which is very impressive. Matrigel-based assay and a tri-dimensional (3D) vascularized model supported the hypothesis very well. However, I have a few comments that need further attention prior to resubmission.
1. In your introduction part, line 50-51 on page 2, you mentioned those references reported reduced angiogenic activities. Could you please discuss the difference of your CRISPR-Cas9 system from the siRNA system or RNF213 KD zebrafish & mouse models in those papers? Why did you get difference results?
By the way, in reference 15, overexpression of RNF213 R4810K downregulated Securin, inhibited angiogenic activity (36.0 ± 16.9%). This is consistent with MMD pathophysiological features. You might reduce your tone here.
2. Line 167, Page 4, I think the abbreviation of ‘polyvinylidene fluoride blotting membrane’ is ‘PVDF’.
3. What is the status of the cells in figure 1, proliferation or confluency?
In figure 1a, the right border of your western blot picture is missing, please add it. Figure 1d, the fluorescence value of X, Y-axis is missing. For cell-cycle analysis, if you could provide the area parameter histogram, that is better.
4. You took the images and analyzed the tube formation assay at 4 hours post seeding cells. From my experience, the tube started to form at 4-6 hours. 4 hours is a little early. What’s the tube formation status at 16 hours, 24hours and 48 hours? What about other indexes indicating tube formation, like number of nodes and number of branches?
5. Figure 4a, from the duplicate spots, I can say TIMP-4 and uPA is reduced in RNF213-/- group. But the mean pixel density is opposite. Could you explain why? Thanks.
6. The IF in figure 6 indicated increased VEGFR2 in RNF213-/-. However, in figure 4d, VEGFR2 has no change. Please explain.
7. What are the results of scratch wound assay and tube formation assay if you over-expressed RNF213 in hCMEC\D3 cells? Could you overturn or reduce cell migration and tube-like formation if you over-expressed RNF213 in your hCMEC\D3-RNF213-/- cells?
8. Could you see the same phenomenon in other endothelial cell lines?
Author Response
We thank the Reviewers for their positive and insightful comments on this manuscript.
-In this manuscript, authors generated an in vitro model of Moyamoya disease (MMD) using the CRISPR-Cas9 system.
-In this model, knockout of RNF213 promotes angiogenesis of cerebral microvascular endothelial cells in a cellular state dependent manner, which is very impressive.
-Matrigel-based assay and a tri-dimensional (3D) vascularized model supported the hypothesis very well.
Comments:
- - “In your introduction part, line 50-51 on page 2, you mentioned those references reported reduced angiogenic activities. Could you please discuss the difference of your CRISPR-Cas9 system from the siRNA system or RNF213 KD zebrafish & mouse models in those papers? Why did you get difference results?”
We agree with the reviewer that such information is essential to discuss and we, accordingly, include a whole section on this very subject in the discussion as suggested.
- “By the way, in reference 15, overexpression of RNF213 R4810K downregulated Securin, inhibited angiogenic activity (36.0 ± 16.9%). This is consistent with MMD pathophysiological features. You might reduce your tone here.”
We are deeply sorry about that and we agree that our level of excitation here could have been misinterpret. We therefore revisited this part of the introduction which is now better reflecting the literature.
- - “Line 167, Page 4, I think the abbreviation of ‘polyvinylidene fluoride blotting membrane’ is ‘PVDF’.”
Thanks for noticing this unfortunate mistake, which has been corrected in the new version.
- - “What is the status of the cells in figure 1, proliferation or confluency?”.
Sorry for this confusion. The cells were at confluency for panel 1a in order to harvest higher amount of total protein, since we wanted to show loss of RNF213 protein expression. All the other panels, included in figure 1 (b-c-d-e-f), showed results using cells in proliferation as the main goal here was to measure proliferation and cell division rate. This information is now properly provided in the revised version of the manuscript.
- “In figure 1a, the right border of your western blot picture is missing, please add it. Figure 1d, the fluorescence value of X, Y-axis is missing. For cell-cycle analysis, if you could provide the area parameter histogram, that is better.”
Thank you again for noticing these unfortunate mistakes, it must be again a problem that occurred during figure upload. We also provide the area parameter as suggested in a supplementary figure S2.
- - “You took the images and analyzed the tube formation assay at 4 hours post seeding cells. From my experience, the tube started to form at 4-6 hours. 4 hours is a little early. What’s the tube formation status at 16 hours, 24hours and 48 hours? What about other indexes indicating tube formation, like number of nodes and number of branches?”.
We actually already performed time-course tube-formation experiment and found that the networks were dissembled/destroyed by 16 hours, as shown in the figure below. We provided the results only for the 4 hours condition as it was the best working condition in our hand.
We also added as suggested more indexed parameters of the tube formation assays in supplementary figures (S3 and S8). Data were reported in fold change relative to the WT control to simplify the representation.
- 5. - “Figure 4a, from the duplicate spots, I can say TIMP-4 and uPA is reduced in RNF213-/- But the mean pixel density is opposite. Could you explain why? Thanks.”
Thanks to the reviewer for noticing this seemingly inconsistence. This can be explained by the fact that the mean pixel densities were normalised with the number of cells present in the well. hCMEC/D3-RNF213-/- are larger cells than the WT control (2.34-fold more cells at confluency), as we have previously published in Stroke earlier this year (Roy et al., 2022) and could have led here this false impression following data normalisation. Since TIMP-4 and uPA are not essential within the parameter of our study, we have decided to remove the data from the principal figure to avoid any confusion, but we indeed kept them in the supplementary figure (S6).
Roy, V.; Ross, J.P.; Pépin, R.; Cortez Ghio, S.; Brodeur, A.; Touzel Deschênes, L.; Le-Bel, G.; Phillips, D.E.; Milot, G.; Dion, P.A.; et al. Moyamoya Disease Susceptibility Gene RNF213 Regulates Endothelial Barrier Function. Stroke 2022, 53, 1263-1275, doi:10.1161/strokeaha.120.032691.
- - “The IF in figure 6 indicated increased VEGFR2 in RNF213-/-. However, in figure 4d, VEGFR2 has no change. Please explain.”
We agree with the reviewer that the IF may give the impression of an overexpression of VEGFR2 in hCMEC/D3-RNF213-/-. However, this observation could be explained by the fact that KO endothelial cells formed thinner and more abundant microvessels. It has to be noted here that the presented IF data have not been normalised. In general, it is not recommended to quantify such unnormalized IF signal.
- - “What are the results of scratch wound assay and tube formation assay if you over-expressed RNF213 in hCMEC\D3 cells? Could you overturn or reduce cell migration and tube-like formation if you over-expressed RNF213 in your hCMEC\D3-RNF213-/- cells?”
This approach could also be really interesting. We in fact tried to do it without success. It has to be noted here that RNF213 is encode by a relatively large gene (cDNA = 15 Kb) and transfection/transduction of such a vector is extremely challenging.
- - “Could you see the same phenomenon in other endothelial cell lines?”
Thanks for the question, which is the subject of ongoing discussion within our group. We have previously published that RNF213 is more expressed in hCMEC/D3 than HUVEC or HMVEC (Roy et al., 2022), suggesting a key role for RNF213 in cerebral endothelial cells. It is therefore a question we would like to explore in the future by performing genome edition as we did here, using different endothelial cells.
Roy, V.; Ross, J.P.; Pépin, R.; Cortez Ghio, S.; Brodeur, A.; Touzel Deschênes, L.; Le-Bel, G.; Phillips, D.E.; Milot, G.; Dion, P.A.; et al. Moyamoya Disease Susceptibility Gene RNF213 Regulates Endothelial Barrier Function. Stroke 2022, 53, 1263-1275, doi:10.1161/strokeaha.120.032691.

Round 2
Reviewer 2 Report
The authors have addressed my concerns in this revision. There is only one minor suggestion. Please add the WB bands in Figure 1a. Thanks.
